# Rethinking Exploration Through Context for RL

*Abstract*—Reinforcement learning has become the dominant post-training paradigm for foundation models, but self-improvement is bottlenecked by exploration. Standard action-space perturbations induce only local exploration, whereas many tasks demand strategies globally different from the current policy — especially when fine-tuning generalist policies on hard exploration tasks with near-zero initial success. We propose perturbing task descriptions instead of actions: prompt perturbation changes what task the model is told to do, steering it toward qualitatively distinct strategies. We cast this as posterior sampling in prompt space, where a distribution over prompts implicitly defines a distribution over policies through the pretrained language prior. To update this distribution without gradient training, we use a vision-language model as both a sampler of plausible prompts and a reasoner that shifts probability toward prompts that elicit success, given observed trajectories. We call the resulting algorithm Prompt-Driven Exploration (PDE). On hard exploration tasks with near-zero initial success, PDE attains higher success rates with far fewer environment interactions than action-space exploration.

*Index Terms*—Reinforcement learning, Vision language action models, Exploration

## I. Introduction

Reinforcement learning (RL) has become a dominant post-training paradigm for foundation models because it enables scalable self-improvement beyond supervised learning. For instance, RL fine-tuning has unlocked strong reasoning and math capabilities in large language models (LLMs) [1], [2], and direct alignment with human preferences for large diffusion-based text-to-image models [3]. However, self-improvement is bottlenecked by exploration: policies can only surpass their current behavior by trying strategies they do not already favor. All exploration methods reduce to the same primitive: perturb some component of the model to produce new behaviors.

Standard practice places the responsibility of this perturbation on the sampled actions [4]–[6]. However, action perturbations only induce local exploration [7], [8]; the set of behaviors reachable by step-wise noise rapidly shrinks with the horizon and action dimension, yet difficult tasks often demand strategies globally different from the current policy. Without a strong warm start, RL rarely finds a good policy — particularly in VLA fine-tuning on manipulation, where state-of-the-art models often start at near-zero success rates [9], [10].

If local exploration is insufficient, how do we perturb globally to reach distinct behaviors? Intuitively, local perturbation changes how a model performs a task, whereas global perturbations should change what task the model is told to do. Consider a robot learning to open a microwave: action jittering nudges the arm's trajectory, but the robot keeps trying the same strategy (eg. reach the handle, pull the door). Global exploration instead pushes the robot toward qualitatively different strategies, such as approaching the handle from a different angle or opening the door by other means. This suggests a concrete mechanism: perturbing the task descriptions themselves.

Perturbing how a task is described connects naturally to context engineering and prompt optimization in foundation models [11], [12], where the phrasing and context of a prompt are known to significantly shape model behavior. Figure 1 illustrates this for VLAs: varying the prompt for a fixed task produces qualitatively distinct trajectories, some of which succeed where the original prompt fails. Prompts are therefore a viable handle for global exploration — the remaining question is how to discover prompts that lead to successful trajectories.

To answer this question, we draw inspiration from posterior sampling-based RL [7], [13]–[15], which maintains a distribution over policies and explores by sampling a policy and rolling it out. Our key observation is that prompts offer a compact, structured surrogate for such a distribution: a VLA defines a family of conditional policies indexed by its prompt, so a distribution over prompts implicitly defines a distribution over policies, with support and structure already shaped by the pretrained language prior. Posterior sampling in policy space thus reduces to posterior sampling in prompt space — we sample a prompt, roll out the induced policy, and update the prompt distribution based on what the trajectory reveals. Performing this update exactly is still difficult, as prompts live in a combinatorial, unbounded space of natural language, where neither the prior nor the likelihood admits a tractable closed form.

Recent advances in vision-language models (VLMs) make prompt optimization for VLAs tractable. A VLM is itself a distribution over language [16]–[18], from which diverse and plausible prompts can be sampled to command the VLA. A VLM is also a multimodal reasoner [19]–[21], capable of observing the trajectories these prompts produce and proposing how the distribution should shift toward prompts that yield better performance. We exploit both capabilities to perform the posterior update without any gradient training, letting a VLM condition on the collected interaction history to update the prompt distribution. This reflects recent evidence that in-context learning in large pretrained models behaves as implicit Bayesian inference over latent task variables [22]–[24], making a VLM a natural amortized approximation to the posterior update. We call the resulting algorithm **Prompt-Driven Exploration (PDE)**: a VLM iteratively updates a distribution over prompts from observed trajectories, and each rollout is driven by a prompt sampled from it — turning

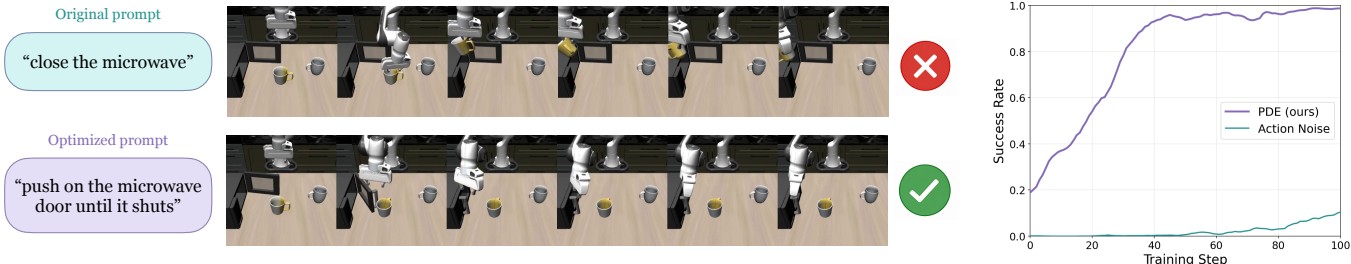

Fig. 1. **Left:** The SFT checkpoint fails under the original prompt "close the microwave"—the robot grasps the mug instead of pushing the door (top row). Our prompt optimization discovers an alternative prompt, "push on the microwave door until it shuts", that redirects the policy to the correct contact point and action, achieving nonzero success without any weight updates (bottom row). **Right:** RL with our prompt driven exploration (PDE) bootstraps from 0% to ~98% success on the *original* prompt, while standard action-noise exploration remains low success rate throughout 100-step training.

prompt optimization into a principled exploration mechanism for RL post-training of foundation-model policies.

We evaluate PDE on LIBERO [25] and LIBERO-PRO [26], using VLAs trained on a fraction of the demonstrations so that initial success rates are near zero. Here the policy has no success to refine, and action-space perturbation rarely discovers one by chance. Across tasks of varying difficulty, PDE reaches higher success rates with far fewer environment interactions, making progress on tasks action-space exploration cannot solve. Analysis shows that the VLMs reliably shift the prompt distribution toward prompts that elicit success, closing the theoretical loop motivating our design.

## II. RELATED WORKS

**RL post-training for VLAs.** A growing body of work fine-tunes pretrained VLAs under sparse binary task success. The most direct approach—on-policy RL end-to-end through the full VLA—has been explored but struggles to scale reliably given the size of the policy and the scarcity of reward signal [27]–[29]. Subsequent work therefore reduces what RL must update: freezing the backbone and training only a small module on top, such as an RL token with actor-critic head [30] or residual off-policy actors later distilled back into the base policy [31]; or moving optimization into the lower-dimensional latent space of a pretrained diffusion policy [32]. These approaches change how the policy is updated; we change what the policy is asked to do by varying the prompt.

**Foundation reward models.** A complementary line densifies the reward itself, training text-conditioned models that score progress toward a specification. One family learns such scores as value functions under a minimum-time goal-reaching assumption [33], [34]; another pseudo-labels demonstration videos with interpolated progress, either globally or within segmented subtasks [35]–[37]; a third learns from preferences [38]. These reshape the reward landscape for a given rollout distribution, while our prompt-space exploration changes the rollout distribution itself; the two are compatible.

**Prompt optimization.** Prior work on LLMs has shown that prompts can be iteratively refined to exceed human performance by using language models as prompt optimizers [16],

[17]. Prior work also explores discrete prompt optimization for VLMs. IPO [39] extends OPRO-style prompt optimization to VLMs and grounds the prompt generation in visual context via a multimodal model. There has also been previous work on soft prompt tuning for VLMs, CoOp [40] optimizes continuous context embeddings via gradient descent rather than natural language. CoVer-VLA [41] uses a trained contrastive verifier to optimizer language instructions to a VLA at test time. While this is the closest related work to ours, our approach uses prompt diversity as an exploration mechanism during training.

**RL exploration strategies.** Beyond action noise, prior work has pursued exploration bonuses—count-based rewards [42], curiosity-driven intrinsic motivation [43], and entropy regularization [6]—and parameter-space noise [44], which perturbs model weights directly. These have seen limited adoption in VLA post-training, where auxiliary density or forward models over high-dimensional visual observations are costly to train alongside billion-parameter policies. Our approach is complementary: rather than perturbing the policy's actions or weights, we explore in the space of task specifications by treating the prompt as a variable, which better leverages the language-conditioning capabilities already present in pretrained VLAs.

## III. PRELIMINARIES

We fine-tune a VLA policy $\pi_\theta(a \mid o, p)$ parameterized by $\theta$ on a task distribution $\mathcal{G}$, sampling a task $g \sim \mathcal{G}$ at each training iteration. Actions $a \in \mathcal{A}$ are conditioned on an RGB image $o \in \mathcal{O}$ and a natural-language prompt $p \in \mathcal{P}$. Each task $g$ is defined by a completion criterion that depends on $g$ alone, not on how it is described in language, and yields a binary reward $r \in \{0, 1\}$ given iff the criterion is met within horizon $H$.

Each task $g$ is paired with a canonical deployment prompt $p_0^g$ (e.g., "open the microwave"). We call two prompts equivalent for $g$ if they describe the same task, and write $\mathcal{P}_g \subseteq \mathcal{P}$ for the set of prompts equivalent to $p_0^g$—rephrasings, different granularities, or sub-goal decompositions (e.g., "pull the microwave door open by its handle"). All prompts in $\mathcal{P}_g$ share the same reward function $r$, but may induce different policies $\pi_p := \pi_\theta(\cdot \mid \cdot, p)$ because the VLA is sensitive

to prompt wording. A rollout under $p$ produces a trajectory $\tau = (o_1, a_1, \ldots, o_H, a_H)$, and we write $J(p) := \mathbb{E}_{\tau \sim \pi_p}[r(\tau)]$.

## IV. METHOD: PROMPT-DRIVEN EXPLORATION (PDE)

**Overview.** Standard exploration perturbs behavior at the action level, producing local deviations around one base policy; the probability of stitching such deviations into a coherent long-horizon success decays geometrically in $H$. PDE instead exploits the one-to-many prompt–task relation: although $p_0^g$ elicits no success, some equivalent prompt $p \in \mathcal{P}_g$ may, and the induced policy $\pi_p$ can supply the gradient signal needed to improve behavior under $p_0^g$. A prompt reconditions the policy for the entire episode, so different prompts induce qualitatively distinct strategies—different approach angles, sub-goal orderings, grasp choices—whose diversity is shaped by the language prior rather than by isotropic noise. PDE realizes this in three parts: (i) we cast prompt search as posterior sampling over policies (Sec. IV-A); (ii) a VLM supervisor plays the role of the posterior, proposing prompts and updating its distribution from rollouts (Sec. IV-B); and (iii) mixture sampling and mixed backpropagation transfer gradient signal from successful prompts back to the deployment prompt $p_0^g$ (Sec. IV-C).

### A. Posterior Sampling over Prompts

**PSRL recap.** For an MDP $M$, write $V^M(\pi) := \mathbb{E}_{\pi, M}\left[\sum_{h=1}^{H} r_h\right]$ for the expected $H$-step return. PSRL [13], [45] maintains a posterior $f_t$ over MDPs and, per episode, samples $M_t \sim f_t$, executes $\pi_t \in \arg\max_{\pi} V^{M_t}(\pi)$, and updates $f_t$ by Bayes' rule. Exploration arises from committing to one self-consistent model per episode rather than step-wise noise.

**Prompt-space PSRL.** With $\theta$ fixed, each prompt induces a distinct policy $\pi_p$, so a distribution $q_t$ over $\mathcal{P}_g$ is a distribution over policies—with support already shaped by the language prior. Each PDE iteration: (1) sample $p_t \sim q_t$; (2) roll out $\pi_{p_t}$, observing $(\tau_t, r(\tau_t))$; (3) update $q_t \to q_{t+1}$; (4) take a policy-gradient step on $\theta$. Three properties make prompt space a favorable substrate for PSRL: $\mathcal{P}$ is natural language, so candidates carry hints a scalar return cannot; step (2) is free, since $\pi_{p_t}$ is fully determined by $p_t$; and the update in step (3) can condition on full trajectories, not just scalars.

### B. VLM Supervisor as Implicit Posterior

$q_t$ has no closed form in the combinatorial space of natural language. We represent it implicitly through a VLM supervisor $\mathcal{V}$ whose in-context buffer stores the history of evaluated prompts and their rollout summaries: sampling from $q_t$ is a query to $\mathcal{V}$, updating $q_t$ is extending $\mathcal{V}$'s context and re-querying. Conditioning $\mathcal{V}$ on $p_0^g$ restricts its proposals to $\mathcal{P}_g$ by construction. This is supported by evidence that in-context learning behaves as implicit Bayesian inference over latent task variables [22]–[24]. Prompting protocols and context compression are in Appendix A.

---

**Algorithm 1** Prompt-Driven Exploration (PDE).

**Require:** VLA $\pi_\theta$; task distribution $\mathcal{G}$; VLM supervisor $\mathcal{V}$; iterations $T$
1: **for** $t = 1, \ldots, T$ **do**
2:     Sample task $g \sim \mathcal{G}$ with deployment prompt $p_0^g$
3:     Sample prompt $p_t \sim \alpha_t \delta_{p_0^g} + (1 - \alpha_t) q_t$
4:     Roll out $\pi_{p_t}$ on $g$; observe $(\tau_t, r(\tau_t))$
5:     $q_{t+1} \leftarrow$ update of $q_t$ via $\mathcal{V}$ using $(p_t, \tau_t, r(\tau_t))$
6:     Update $\theta$ by a PPO step using $\log \pi_{\mathrm{mix}}$
7: **end for**
8: **return** $\pi_\theta$

---

### C. Implementation

We instantiate PDE on top of PPO [5], whose update on $\theta$ is, up to clipping, a gradient step on the importance-weighted surrogate

$$\mathcal{L}^{\mathrm{PPO}}(\theta) = \mathbb{E}_t\left[\rho_t(\theta)\,\hat{A}_t\right], \qquad \rho_t(\theta) = \frac{\pi_\theta(a_t \mid o_t, p)}{\pi_{\mathrm{old}}(a_t \mid o_t, p)}, \quad (1)$$

where $(o_t, a_t)$ is the observation–action pair at step $t$ of a rollout collected under prompt $p$, $\pi_{\mathrm{old}}$ denotes the policy at collection time, and $\hat{A}_t$ is an advantage estimate. The prompt appearing in $\pi_\theta$ is precisely the conditional that each gradient step pushes upward—so training on rollouts collected under $p \sim q_t$ improves $\pi_\theta(\cdot \mid \cdot, p)$, not $\pi_\theta(\cdot \mid \cdot, p_0^g)$, yet deployment is evaluated under $p_0^g$. We close this gap with two modifications.

**Mixture sampling.** Each rollout draws its prompt from $p \sim \alpha_t \delta_{p_0^g} + (1 - \alpha_t) q_t$, with $\alpha_t$ annealed from a small floor toward 1 as on-prompt success rises. Rollouts under $q_t$ supply gradient signal; rollouts under $p_0^g$ align training with deployment.

**Mixed backprop.** Sampling alone is insufficient: a rollout drawn under $p$ still yields a gradient that targets only $\pi_\theta(\cdot \mid \cdot, p)$. In practice $\rho_t$ is computed as $\exp(\log \pi_\theta(a_t \mid o_t, p) - \log \pi_{\mathrm{old}}(a_t \mid o_t, p))$, so it is the current-policy log-probability that the optimizer ultimately sees. For each batch we therefore run two forward passes of the policy—one under the sampled $p$, one under $p_0^g$—and average their log-probabilities,

$$\log \pi_{\mathrm{mix}}(a \mid o) = \tfrac{1}{2} \log \pi_\theta(a \mid o, p_0^g) + \tfrac{1}{2} \log \pi_\theta(a \mid o, p), \quad (2)$$

and substitute $\log \pi_{\mathrm{mix}}$ for $\log \pi_\theta(a_t \mid o_t, p)$ in this ratio. Averaging in log-space couples the two prompts multiplicatively: each update is rewarded only when the action is probable under *both* $p$ and $p_0^g$, suppressing gradients that would raise $\pi_\theta(\cdot \mid \cdot, p)$ at the expense of $\pi_\theta(\cdot \mid \cdot, p_0^g)$. Algorithm 1 summarizes the full procedure; the $\alpha_t$ schedule, the two-regime separation of $q_t$ and $\theta$ updates, and hyperparameters are in Appendix A.

## V. EXPERIMENTS

We evaluate our method on the LIBERO benchmark suite, addressing three questions: (i) Does VLM-driven prompt optimization discover prompts that yield nonzero success on tasks where the SFT checkpoint achieves 0%? (ii) Does RL with a prompt curriculum bootstrap a deployable policy under

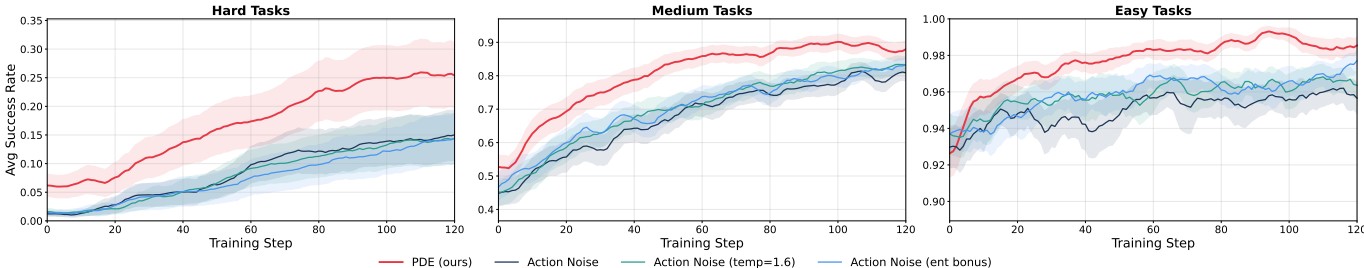

Fig. 2. Aggregated training curves grouped by task difficulty based on initial PPO baseline success rate. **Left:** Hard tasks (init SR $< 0.1$, 46 tasks). **Middle:** Medium tasks ($0.1 \leq$ init SR $< 0.8$, 38 tasks). **Right:** Easy tasks (init SR $\geq 0.8$, 36 tasks). Shaded regions indicate standard error of the mean across tasks. The larger standard error on hard tasks reflects high variance between tasks that remain near zero throughout training and those that eventually improve.

the *original* prompt? (iii) Do these gains generalize across a diverse benchmark of out-of-distribution tasks? We answer (i) and (ii) with an illustrative case study on a single task whose failure mode is easy to diagnose (Section B2), then scale to the full LIBERO-PRO benchmark—120 tasks spanning four suites and three perturbation types (Section V-B). Section B5 ablates the contribution of the prompt curriculum, the adaptive schedule, and the mixed backpropagation scheme.

### A. Setup

**Environment and model.** All experiments use Pi0.5 as the base model, trained with supervised fine tuning (SFT) on the four LIBERO suites (goal, spatial, object, 10) using only 40 trajectories in total (10 per suite). We freeze this checkpoint for offline prompt optimization and use it as the initialization for all RL fine-tuning. Hyperparameters and full implementation details are in Appendix B.

**Metric.** We report `success_once`, the binary episode-level success metric used in the RLinf evaluation suite, averaged over 240 evaluation environments per task.

**Baselines.** We compare against four baselines, all initialized from the same SFT checkpoint and trained on the original task prompt.

- **Action Noise**: standard on-policy fine-tuning with the binary task reward, training-time temperature 1.0, and no entropy bonus.
- **Action Noise + high temperature**: identical to PPO but with training-time temperature raised to 1.6 to encourage action-space exploration.
- **Action Noise + entropy bonus**: identical to PPO but with an entropy bonus term of coefficient 0.01 added to the policy-gradient objective.

The high-temperature and entropy-bonus variants test whether traditional action-space exploration—injecting noise directly into the policy's action distribution—is sufficient to escape the zero-reward regime. We use the same PPO optimizer settings and rollout budget across all four baselines and our method.

### B. Benchmark Results on LIBERO-PRO

**Setup.** We benchmark on LIBERO-PRO [26], whose perturbed task variants span initial success rates from $0\%$ to $> 90\%$, providing a natural mixture of tasks that can and

cannot be learned with action-noise exploration alone. We evaluate on three of its five perturbation types: *task* (changed goal instruction), *swap* (permuted object positions), and *object* (visually similar object replacement). We exclude *language* perturbation (confounded with our prompt modification) and *environment* perturbation (BDDL files not publicly released). Each (suite, perturbation) combination contains 10 tasks, giving $4 \times 3 \times 10 = 120$ tasks total. We train one policy per combination (jointly over its 10 tasks) and report success rates at training step 120.

**Does PDE improve over standard action noise?** Since our method is designed primarily to help when the base policy provides little or no reward signal, we group the 120 tasks into three difficulty tiers by SFT success rate to test whether the gains concentrate on the hardest tasks as predicted: *hard* (init SR $< 10\%$; 47 tasks), *medium* (10–80%; 36 tasks), and *easy* ($\geq 80\%$; 37 tasks). Table II and Figure 2 report success rates aggregated by tier. Our method outperforms all three action-noise baselines across every difficulty tier, with the largest gap on the *hard* tier (60% relative improvement), where many tasks have zero initial success and action-noise methods receive no learning signal. This is consistent with the microwave case study (Section B2): prompt-space exploration discovers language groundings that unlock nonzero performance on tasks where the SFT checkpoint has overfit to a wrong motor program, precisely the regime in which action-space perturbations are ineffective.

## VI. CONCLUSION

We introduced Prompt-Driven Exploration (PDE), which reframes RL exploration as posterior sampling in prompt space, using a VLM to update a distribution over prompts from observed trajectories. On LIBERO and LIBERO-PRO, PDE solves tasks action-space exploration cannot, with far fewer interactions. The framework extends to any prompt-conditioned policy — LLM agents, tool-use systems, reasoning models — and prompts are only one of several structured surrogates through which a posterior over behavior could be expressed. More broadly, for foundation-model policies, the most useful exploration signal often lives not in the action space the policy outputs but in the context space it conditions on.

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

# APPENDIX

## A. Implementation Details

*1) VLM Supervisor as an Implicit Posterior:* We represent $q_t$ implicitly through a VLM supervisor $\mathcal{V}$ whose in-context buffer encodes the procedure's history. At iteration $t$, let

$$\mathcal{H}_t = \{(p_i, c_i, s_i)\}_{i<t}, \qquad (3)$$

where $s_i \in [0,1]$ is the empirical success rate of prompt $p_i$ over $N$ rollouts and $c_i$ is a one-sentence natural-language summary of the rollout behavior. Let $\mathcal{P}_t \subseteq \{p_i\}_{i<t}$ denote the running pool of prompts whose empirical success rate exceeds a threshold $\eta$. We set

$$q_t(p) \equiv \mathcal{V}(p \mid p_0^g, \mathcal{P}_t, \mathcal{H}_t), \qquad (4)$$

conditioning on the deployment prompt $p_0^g$ so that proposals lie in $\mathcal{P}_g$ by construction. Sampling from $q_t$ reduces to a query for new candidate prompts conditioned on $(p_0^g, \mathcal{P}_t, \mathcal{H}_t)$; updating $q_t$ reduces to appending the latest evaluation to $\mathcal{H}_t$ and re-querying at the next iteration.

*a) Summary generation:* Immediately after rolling out $p_i$ for $N$ episodes, we issue a one-off query to $\mathcal{V}$ on the rollout videos asking for a single sentence describing what the policy attempted and where it failed. Only the sentence $c_i$—not the videos themselves—is retained in $\mathcal{H}_t$, so the supervisor's context grows linearly in the number of evaluated prompts rather than in the number of frames. The exact prompt used for this query is given in Appendix A5.

*b) Pool admission:* At each iteration we evaluate $K$ candidate prompts. A candidate $p$ with success rate $s \geq \eta$ is admitted to $\mathcal{P}_t$; others are recorded in $\mathcal{H}_t$ but do not enter the pool. Candidates with $s = 1$ are kept in $\mathcal{P}_t$ without further refinement.

*2) Two-Regime Schedule:* The main text notes that updates to $q_t$ and $\theta$ must run on separate schedules, since a moving $\theta$ changes which prompts count as "good" and renders the posterior over prompts non-stationary. We instantiate this with the simplest possible schedule: a single trajectory with two regimes.

- **Regime A (pool building, $t \leq T_0$):** $\theta$ is frozen. Each iteration samples $K$ candidate prompts from $\mathcal{V}$, evaluates each for $N$ rollouts, and updates $\mathcal{H}_t, \mathcal{P}_t$.
- **Regime B (policy optimization, $t > T_0$):** $\mathcal{P}_t$ is frozen at $\mathcal{P}_{T_0}$ and $q_t$ ceases to update. Rollouts are collected under a prompt mixture (Section A3) and used for PPO updates on $\theta$.

The transition from A to B is triggered either when $|\mathcal{P}_t| \geq M_{\min}$ or when $t = T_0$, whichever comes first. This ensures entering Regime B with a non-empty pool of success-eliciting prompts.

*3) Anchoring Policy Updates to the Deployment Prompt:* In Regime B, two modifications keep $\theta$ aligned with the deployment prompt $p_0^g$ while still benefiting from the reward signal

available under $\mathcal{P}_{T_0}$. This section expands the $\alpha_t$ schedule and the mixed-backprop loss sketched in Section IV-C.

*a) Mixture sampling.:* Each rollout in Regime B draws its prompt from

$$p \sim \alpha_t \cdot \delta_{p_0^g} + (1 - \alpha_t) \cdot \mathrm{Uniform}(\mathcal{P}_{T_0}), \qquad (5)$$

where $\alpha_t \in [\alpha_{\min}, 1]$. This is the concrete instantiation of the $q_t$ in Section IV-C: once Regime B begins, $q_t$ reduces to a uniform distribution over the frozen pool $\mathcal{P}_{T_0}$. Rollouts under $\mathcal{P}_{T_0}$ supply gradient signal; rollouts under $p_0^g$ keep the training distribution aligned with deployment.

*b) Adaptive coefficient.:* Let $s_t^{(p_0^g)} \in [0, 1]$ be the empirical success rate of rollouts conditioned on $p_0^g$ at step $t$. We track an exponential moving average and set

$$\bar{s}_t = \beta\, s_t^{(p_0^g)} + (1 - \beta)\, \bar{s}_{t-1}, \qquad (6)$$

$$\alpha_t = \mathrm{clip}\Big(\frac{\bar{s}_t}{c},\ \alpha_{\min},\ 1\Big), \qquad (7)$$

with smoothing factor $\beta$, consolidation target $c \in (0, 1)$, and floor $\alpha_{\min} > 0$. When $\bar{s}_t \approx 0$, $\alpha_t = \alpha_{\min}$ and most rollouts come from $\mathcal{P}_{T_0}$, where the gradient signal is dense. As $\bar{s}_t$ rises, $\alpha_t$ anneals toward 1 and training consolidates onto $p_0^g$; full consolidation is triggered at $\bar{s}_t = c$ rather than at perfect performance. The floor guarantees nonzero gradient signal on $p_0^g$ throughout training.

*c) Mixed backpropagation.:* Because the PPO ratio $\rho_t = \pi_\theta(a_t \mid o_t, p)/\pi_{\mathrm{old}}(a_t \mid o_t, p)$ is implemented as $\exp(\log \pi_\theta(a_t \mid o_t, p) - \log \pi_{\mathrm{old}}(a_t \mid o_t, p))$, the current-policy log-probability is the quantity the optimizer ultimately sees, and a gradient computed only under the sampled prompt $p$ pushes the policy toward $\pi_\theta(\cdot \mid \cdot, p)$ rather than toward $\pi_\theta(\cdot \mid \cdot, p_0^g)$. For each micro-batch we therefore run two forward passes—one under $p$ and one under $p_0^g$—and substitute the averaged log-probability

$$\log \pi_{\mathrm{mix}}(a \mid o) = \tfrac{1}{2} \log \pi_\theta(a \mid o, p_0^g) + \tfrac{1}{2} \log \pi_\theta(a \mid o, p) \qquad (8)$$

for $\log \pi_\theta(a_t \mid o_t, p)$ in $\rho_t$, evaluated against the old rollout log-probabilities recorded at collection. Averaging in log-space couples the two prompts multiplicatively: an update is rewarded only when the action is probable under *both* $p$ and $p_0^g$, suppressing gradients that would raise $\pi_\theta(\cdot \mid \cdot, p)$ at the expense of $\pi_\theta(\cdot \mid \cdot, p_0^g)$. Gradients flow through both forward passes, so each update simultaneously improves the policy under the sampled prompt and under the deployment prompt.

*4) Hyperparameters:* Table I lists all hyperparameters used in our experiments. Unless stated otherwise, values are shared across LIBERO and LIBERO-PRO.

*5) Supervisor Prompt Templates:* We reproduce the three prompt templates used to query $\mathcal{V}$: (i) the candidate-generation template, which takes $(p_0^g, \mathcal{P}_t, \mathcal{H}_t)$ as input and returns $K$ proposed prompts; (ii) the rollout-summarization template, which takes $N$ rollout videos of a single prompt as input and returns a one-sentence summary $c$; (iii) the system prompt that frames the supervisor's role. Each template is given verbatim below, with placeholders in {curly braces}.

TABLE I
HYPERPARAMETERS FOR PDE. VALUES IN THE TOP TWO BLOCKS ARE SPECIFIC TO PDE; THE BOTTOM BLOCK LISTS THE STANDARD PPO HYPERPARAMETERS USED FOR POLICY-GRADIENT UPDATES IN REGIME B.

| Symbol | Description | Value |
|---|---|---|
| *Posterior-sampling loop* | | |
| $T_0$ | Pool-building iterations | 10 |
| $K$ | Candidate prompts per iteration | 5 |
| $N$ | Rollouts per candidate | 10 |
| $\eta$ | Pool admission threshold | $> 0$ |
| *Deployment-prompt anchoring* | | |
| $\alpha_{\min}$ | Floor on deployment-prompt mixture weight | 0.05 |
| $\beta$ | EMA smoothing factor | 0.3 |
| $c$ | Consolidation target | 0.5 |
| *PPO* | | |
| $\epsilon$ | Clip range | 0.2 |
| $\eta_{\mathrm{lr}}$ | Learning rate | $5 \times 10^{-6}$ |
| $B$ | Batch size | 2048 |
| $\lambda$ | GAE $\lambda$ | 0.95 |
| $\gamma$ | Discount factor | 0.99 |
| $E$ | Epochs per update | 4 |

*a) (i) Candidate generation and rollout summarization:*

```
## Training Step {step} --- Prompt Pool
**GOAL: {task_description}**
**Original prompt**:
"{task_description}" | EMA success:
{orig_rate} [window trend: {trend}]
### Current Pool Prompts and Their
Rollout Videos
**Prompt**: "{prompt_i}" | EMA success:
{ema_i}
{rollout video or frames for prompt_i}
... (repeated for each prompt in the
pool)
### Your Task
1. **Analyze each video**: What is
the robot doing under each prompt?
Which prompts produce the best behavior
toward the goal?
2. **Summarize each prompt's effect**
in one line.
3. **Generate 1 NEW prompt** to add to
the pool:
- Address failure modes you observed
- Be semantically diverse from existing
prompts
- Try different phrasings, action
verbs, spatial references
Return ONLY valid JSON:
{
"new_prompts": ["new prompt"],
"summaries": {"exact prompt text":
"one-line behavior summary", ...},
"analysis": "brief reasoning about what
to try next"
}
```

*b) (ii) System prompt:*

```
You are an expert in robotic
manipulation and prompt engineering
for vision-language-action models.
Each prompt is accompanied by
{frames_per_video} frames from
a robot rollout video, shown in
temporal order (Frame 1 = start, Frame
{frames_per_video} = end). Trace the
robot's movement across frames to
understand what it does over time.
Your tasks:
1. Analyze the rollout videos --- what
is the robot doing? How close to the
goal?
2. Identify failure modes (wrong
object, wrong location, failed grasp,
etc.)
3. Analyze how different prompts affect
the robot's behavior
4. Generate 1 NEW prompt to add to the
pool
The GOAL is: "{task_description}"
The robot is controlled by a
vision-language-action model (pi0.5).
Guidelines for new prompts:
- Be clear and specific about the
object and target location
- Use natural language similar to
LIBERO benchmark conventions
- Try different levels of specificity,
action verbs, spatial references
- Keep prompts concise (5--15 words
typically)
```

### B. Experiment Details

*1) PDE implementation:* The prompt optimization loop uses Qwen3-VL-235B as the VLM supervisor, with $T = 10$ iterations, $K = 5$ candidate prompts per iteration, and $N = 10$ rollouts per candidate. The RL stage uses PPO with the prompt curriculum described in Section IV-C: at each step, the prompt is sampled from a mixture of the original prompt $p_0$ and the discovered pool $\mathcal{P}_T$ ($|\mathcal{P}_T| = 5$), and the mixing coefficient $\alpha_t = \text{clip}(\bar{s}_t/0.5, 0.05, 1.0)$ is computed *per task* from an EMA of the original-prompt success rate. The PPO gradient is computed under both prompts simultaneously via the mixed-backprop scheme described in Section IV-C.

*2) Illustrative Experiment: Close the Microwave:* **Setup.** We choose the LIBERO-90 task KITCHEN_SCENE6_close_the_microwave for case study (Figure 1, left). The task is specified by the original prompt $p_0$ = "close the microwave". The SFT checkpoint has 0% success rate under $p_0$, and as Figure 1 (right) shows, action-noise baselines only reach $\sim 10\%$ near step 100, confirming that perturbing actions around the wrong motor program is an extremely slow path to success on this task.

**Why does the SFT checkpoint fail?** Inspection of rollout videos reveals a consistent failure mode: the arm grasps the yellow-handled mug, carries it toward the microwave, and stalls—never contacting the door. The SFT data contains a visually identical task, "put the yellow and white mug in the microwave and close it"; the shared scene and the shared word "microwave" cause the policy to execute that learned trajectory regardless of the actual prompt. The model has learned a visual-scene-to-trajectory mapping rather than language-conditioned control, and action-space noise cannot fix this: perturbing actions around the wrong motor program yields the same failure.

**Can prompt optimization find prompts with nonzero success?** Running our offline prompt optimization loop on the SFT checkpoint discovers 10–12 unique prompts with nonzero success per run (across 84 cumulative rollouts each; Figure X). All nonzero prompts achieve 33% success rate (1 of 3 trials). Across runs, the 20 unique discovered prompts cluster into three strategies:

- **Explicit contact and action** ("push the microwave door closed", "push the microwave door until it clicks", "swing the microwave door shut"): names the physical interaction directly, redirecting the motor program away from the pick-and-place prior toward a push-and-close behavior.
- **Spatial/generic reference** ("close the black appliance door on the left", "close the kitchen appliance door", "shut the open appliance door"): refers to the microwave by its appearance or location rather than its label. The SFT model has stronger spatial-affordance grounding than object-name grounding for this scene, so the generic reference bypasses the failed object-name cue.
- **Distractor exclusion** ("do not close the cabinet – close microwave", "first move to the microwave, then close the door"): explicitly redirects attention away from the distractor object, directly addressing the observed failure mode.

Notably, none of the discovered prompts exceed 33%—the prompt optimization identifies the *direction* of improvement (contact grounding and distractor avoidance) but cannot overcome the SFT checkpoint's fundamental motor limitation through language alone, motivating the RL fine-tuning stage.

**Can RL with the prompt curriculum bootstrap to high success on the original prompt?** As shown in Figure 1 (right) The action-noise baseline (standard PPO with $p_0$) remains near 0% for the first 60 steps and reaches only $\sim 10\%$ by step 100. Our method starts at $\sim 20\%$—reflecting the nonzero success of the discovered prompt pool—climbs to $\sim 90\%$ by step 40, and plateaus near 98%. The learning curve exhibits three phases driven by the adaptive schedule: (i) $\alpha_t$ sits at its floor and nearly all rollouts come from the curriculum prompts, building transferable motor skill; (ii) the original-prompt success rate rises, $\alpha_t$ climbs, and the rollout mixture shifts toward $p_0$; (iii) $\alpha_t$ saturates and final convergence proceeds under pure original-prompt training. This smooth consolidation is enabled by mixed backpropagation, which trains the policy on $p_0$ throughout the curriculum phase so that no cross-prompt transfer is needed when $\alpha_t \to 1$.

*3) LIBERO-PRO benchmark setup:* We use LIBERO-PRO [26] as our benchmark because its perturbed task variants create exactly the out-of-distribution conditions our method targets: the SFT checkpoint's initial success rates span the full range from 0% to $> 90\%$ across settings, providing

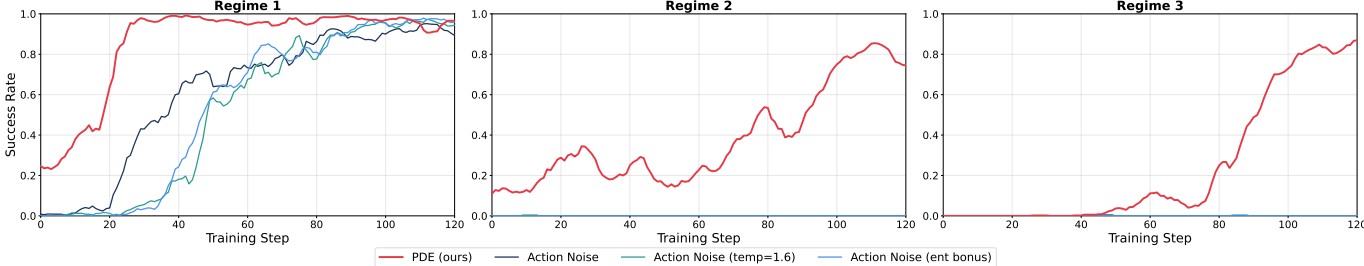

Fig. 4. Per-task training curves for Object Suite with task perturbation, illustrating the three regimes. **Regime 1** (Task 0): both methods eventually converge, but PDE reaches 90% of its peak in roughly half the steps. **Regime 2** (Task 5): the SFT checkpoint has 0% success under the original prompt, but prompt optimization finds a nonzero prompt; standard PPO stays near 0 while PDE climbs steadily. **Regime 3** (Task 2): prompt optimization finds *no* nonzero prompt for this task, yet PDE still achieves nonzero success via *cross-task generalization* from the curriculum of other tasks.

TABLE II
AGGREGATED SUCCESS RATE (%) BY TASK DIFFICULTY ON
LIBERO-PRO

| | hard | medium | easy | all |
|---|---|---|---|---|
| base model | $1.18_{\pm0.37}$ | $45.08_{\pm3.65}$ | $93.98_{\pm0.95}$ | $42.96_{\pm3.70}$ |
| pde (ours) | $\mathbf{27.32}_{\pm6.08}$ | $\mathbf{91.89}_{\pm2.11}$ | $\mathbf{99.46}_{\pm0.44}$ | $\mathbf{68.93}_{\pm3.93}$ |
| action noise | $17.11_{\pm5.02}$ | $86.11_{\pm2.58}$ | $98.70_{\pm0.38}$ | $62.97_{\pm4.00}$ |
| action noise (temp=1.6) | $15.49_{\pm4.74}$ | $88.44_{\pm2.85}$ | $97.73_{\pm0.64}$ | $62.73_{\pm4.04}$ |
| action noise (ent bonus) | $15.23_{\pm4.81}$ | $86.11_{\pm2.59}$ | $98.70_{\pm0.48}$ | $62.23_{\pm4.03}$ |

a natural mixture of tasks that can and cannot be learned with action-noise exploration alone. LIBERO-PRO defines five perturbation types over the four LIBERO task suites; we evaluate on three:

- **Task perturbation** changes the goal instruction so the robot must accomplish a different objective in the same scene—for example, "open the middle drawer" → "open the bottom drawer".
- **Swap perturbation** applies pairwise swaps to initial object positions while keeping the goal unchanged, testing spatial robustness.
- **Object perturbation** replaces an object with a visually similar variant (e.g., a wooden cabinet with a yellow one), testing visual generalization.

We exclude *language perturbation* because our method also modifies the language prompt, which would confound the comparison, and we exclude *environment perturbation* because the corresponding BDDL files have not been publicly released. Each (suite, perturbation) combination contains 10 tasks, giving $4 \times 3 \times 10 = 120$ tasks in total. For RL training, we train one policy per combination, so each run jointly trains on the 10 tasks within that combination. We report the success rate of action-noise baselines and our method at training step 120, along with the SFT checkpoint's initial success rate as a reference.

*4) LIBERO-PRO experiment analysis:* **A finer-grained per-task breakdown reveals three regimes in which PDE helps over standard action noise**:

**Regime 1: faster convergence on nonzero-SR tasks.** On tasks where the SFT checkpoint already achieves nonzero success under the original prompt, both action noise and PDE

eventually converge to comparable peaks, but PDE converges significantly faster, reaching 90% of its peak in roughly half the training steps (Figure 4, left). The EMA curriculum provides denser reward signal early in training, which trains the value function faster and accelerates the entire learning curve.

**Regime 2: prompt-based bootstrapping on zero-SR tasks.** On tasks where the SFT checkpoint has 0% success under the original prompt but offline prompt optimization discovers at least one nonzero prompt, action noise baselines remain stuck near 0 throughout training, while PDE climbs steadily (Figure 4, center). This is the regime the microwave case study illustrates, generalized across the benchmark.

**Regime 3: cross-task generalization for zero-SR tasks where prompt optimization fails.** On some tasks where prompt optimization finds *no* nonzero prompt, PDE still produces nonzero success after RL training (Figure 4, right). The curriculum from the tasks where prompts *were* discovered evidently bootstraps motor skills that transfer to the remaining tasks—a form of cross-task generalization that prompt optimization alone cannot achieve.

*5) Ablation Studies:* **How does the number of discovered prompts scale with the search budget?** Figure 6 plots the cumulative number of nonzero prompts discovered as a function of cumulative rollouts. The discovery rate is approximately linear, with each run finding 9–12 nonzero prompts after 84 rollouts. All discovered prompts achieve 33% success rate (1 of 3 trials), suggesting that a moderate search budget is sufficient to populate the curriculum pool.

**Is the curriculum necessary, or is a single best prompt enough?** The training curves in Figure 5 (left) appear similar because they measure success under the prompt used during training—which for the best-prompt baseline is always the highest-performing discovered prompt. However, at deployment the policy need to execute the *original* task instruction. As shown in Table III, evaluating under the original deployment prompt reveals a substantial gap between the

TABLE III
EVAL SR ON ORIGINAL PROMPT.

| Method | SR |
|---|---|
| Best Prompt (no curr.) | 0.396 |
| Prompt Curriculum | **0.638** |

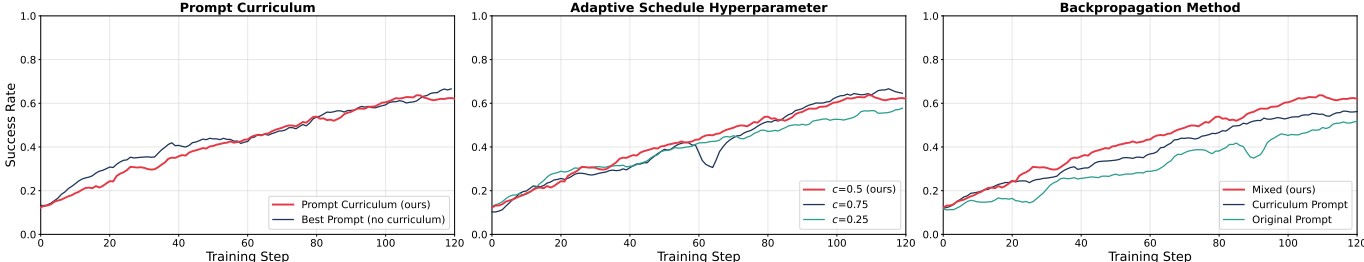

Fig. 5. Ablation studies on the Object suite with task perturbation. **Left:** Prompt curriculum vs. single best prompt. **Middle:** Adaptive schedule hyperparameter $c \in \{0.25, 0.5, 0.75\}$. **Right:** Backpropagation method—mixed (ours), curriculum prompt only, and original prompt only.

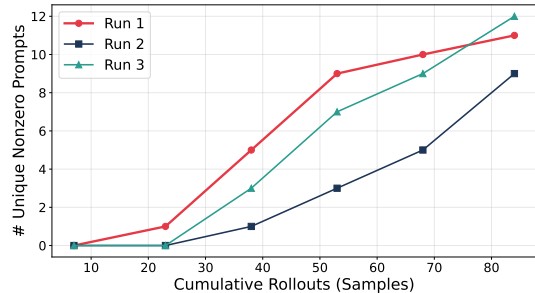

Fig. 6. Cumulative unique nonzero prompts discovered vs. cumulative rollouts across three independent runs on the microwave task.

two methods. Training exclusively on a discovered prompt optimizes the policy for that particular instruction, but the learned behavior does not transfer to the deployment prompt. The curriculum's mixed backpropagation, which alternates between curriculum and original prompts, ensures that the policy is directly optimized for the instruction it will encounter at test time.

**How sensitive is the curriculum to the consolidation target $c$?** Figure 5 (middle) shows that all three settings ($c \in \{0.25, 0.5, 0.75\}$) follow similar trajectories and converge to comparable final success rates (58–65%), indicating that the method is not sensitive to this hyperparameter. We use $c=0.5$ as the default throughout.

**Does mixed backpropagation help?** Figure 5 (right) compares three backpropagation strategies. Original-only backpropagation learns slowest (52% at step 120), as it receives sparse reward signal early in training. Curriculum-only improves faster but plateaus at 56%, since gradient updates are never conditioned on the deployment prompt. Mixed backpropagation achieves the highest final success rate (62%) by combining dense gradient signal from the curriculum prompts with direct optimization on the original prompt.