# OpenReview forum: "Rethinking Exploration Through Context for RL"
_IEEE.org/ICRA/2026/Workshop/Manipulation_Robustness — ICRA 2026_

### Official Review · Reviewer_xM9E · 2026-05-14
**A promising but preliminary systems idea, lacks convincing experiment results**

**Rating:** 7
**Confidence:** 5

**Review:**

**Summary**:

The paper proposes Prompt-Driven Exploration (PDE): instead of exploring by perturbing actions, it explores by perturbing the task prompt given to a pretrained vla model. A vlm supervisor proposes alternative prompts, keeps a pool of prompts that produce some success, and then ppo finetunes the policy using a mixture of the original prompt and discovered new prompts, with a “mixed backpropagation” trick intended to transfer gains back to the deployment prompt. Empirically, the paper shows performance gains on LIBERO-PRO, especially on hard tasks with near-zero initial success

**Weaknesses**:

The benchmark setup seems stacked in favor of the method. The authors explicitly choose a regime where initial success is often near zero. That is the setting where prompt sensitivity and the flaws in the original language conditioned vla are most likely to dominate. So the experiments may be demonstrating that prompt search helps a badly undertrained language-conditioned policy, not that prompt-space exploration is broadly superior in a more generic sense.

The improvements on the hard LIBERO-PRO tasks are only moderate. That is useful, but nowhere near solving hard exploration. The overall score is 68.93 vs 62.97, again a moderate gain rather than a breakthrough. The paper seems to lean on relative percentages and the dramatic case study.

**Questions**:

The environment-interaction comparison seems unfair? If I understand it correctly, the paper says the same rollout budget is used across baselines and PDE, but PDE also has an offline prompt optimization loop with its own rollout evaluations before rl. Those prompt-search episodes are environment interactions too?

---

### Official Review · Reviewer_Hi4v · 2026-05-16
**Promising Prompt-Space Exploration for VLA Post-Training, with an Important Missing Prompt-Pool Baseline**

**Rating:** 7
**Confidence:** 4

**Review:**

## Summary

This paper proposes Prompt-Driven Exploration (PDE), a method for improving sparse-reward RL fine-tuning of VLA policies by exploring over task prompts rather than low-level actions. The key idea is that a prompt-conditioned policy can induce qualitatively different behaviors under semantically equivalent prompts. PDE decomposes exploration into two regimes: first, constructing a pool of semantically equivalent prompts for a frozen policy $\pi_\theta$; and second, using this prompt pool as a curriculum for PPO by exploring actions in the environment under prompt-conditioned policies. By evaluating the final policy on the original deployment prompt, the authors demonstrate that prompt-induced exploration can transfer back to the intended task specification. The approach is well motivated, and the case study in the main text provides clear evidence that prompt-space exploration can change task binding, affordance selection, and strategy-level behavior. However, the experimental suite is missing an important random-prompt-pool baseline, making it difficult to fully disentangle the value of feedback-guided prompt discovery from the benefit of simply training with a larger and more diverse prompt pool.

## Quality

PDE is a novel take on an important bottleneck in VLA post-training. Standard action-space exploration has often been found to be an inefficient strategy with little useful reward signals. The proposed decomposition into prompt discovery, prompt-pool curriculum, adaptive mixture sampling, and mixed backpropagation is practical and well aligned with the problem. My only concern is that the current experimental suite is missing some critical ablations around a random-prompt-pool baseline. This missing ablation raises some concerns about whether the performance gains arise on account of the proposed prompt-exploration strategy, or if they are an artifact of having a larger prompt pool at the time of exploration.

## Clarity

I like the paper presentation. It is written with clarity with the core contributions and problem formulation taking center stage, and being easy to parse. I particularly like the case study on the microwave example.

## Originality

The paper presents a novel take on VLA post-training with RL based on prompt exploration.

## Significance

PDE improves most clearly in the hard-task regime, where the base policy has little initial success and the benefit of better exploration should be most visible. The evaluation on the original deployment prompt is also important, because it shows that the method is not merely finding better test-time prompts, but can transfer prompt-induced behavior back into the intended task specification.

That said, the significance of the prompt-discovery component is somewhat difficult to isolate without a random-prompt-pool baseline. A fixed pool of semantically equivalent prompts generated from $\pi_0$, used with the same curriculum and mixed-backpropagation procedure, would clarify whether the gains come from feedback-guided prompt exploration or from the broader effect of training with a diverse prompt pool.

The absence of real-world evaluation also means the current evidence is best interpreted as a strong simulation proof of concept rather than a complete demonstration of practical VLA post-training.

## Pros

- Clean idea: use prompt space as the exploration space for VLAs.
- Well matched to sparse-reward settings where action noise has little signal.
- The two-regime design is simple and practical.
- Evaluation on the original prompt distinguishes approach from simple prompt optimization.
- The case study clearly shows changes in task binding and affordance selection.

## Cons

- Missing key baseline: a fixed random/one-shot prompt pool from $\pi_0$, used with the same curriculum.
- Best-prompt-only training should be added to main text.
- Simulation-only results make this a proof of concept.

## Overall Assessment

This is a strong workshop paper with a clean idea. The method is practical, the case study is convincing, and the evaluation on the deployment prompt makes the contribution more meaningful than simple test-time prompt optimization.

My main reservation is the missing random/one-shot prompt-pool baseline. Without it, it is hard to tell how much of the gain comes from feedback-guided prompt discovery versus the broader benefit of training with a diverse prompt pool. I would still be positive on the paper, but I would encourage the authors to add this baseline and eventually validate the approach outside simulation.

---

### Decision · Program_Chairs · 2026-05-21

Accept